

# Technical note: RA138 Calcite U–Pb LA-ICP-MS primary reference material

Marcel Guillong[1], Elias Samankassou[2], Inigo A. Müller[2], Dawid Szymanowski[1], Nathan Looser[1], Lorenzo Tavazzani[1], Óscar Merino-Tomé[3], Juan R. Bahamonde[3], Yannick Buret[4], Maria Ovtcharova[2]

[1]Department of Earth Sciences, ETH Zurich, 8092 Zurich, Switzerland
[2]Department of Earth Sciences, University of Geneva, 1205 Geneva, Switzerland
[3]Departamento de Geología, Universidad de Oviedo, Oviedo, Spain.
[4]Imaging and Analysis Centre, Natural History Museum, London, UK.

*Correspondence to*: Marcel Guillong (guillong@erdw.ethz.ch)

**Abstract**

A promising primary reference material for U–Pb LA-ICP-MS carbonate dating is analysed and reported here. The new RM is a botryoidal cement (C1) from sample RA138. The sample was collected in outcrop strata of mid-Carboniferous (Uppermost

Mississippian, upper Serpukhovian) in northern Spain near La Robla, and multiple aliquots have been meticulously prepared for distribution. The RA138 is characterised by variable U/Pb ratios (from ~1 to ~19) and a relatively high and homogeneous U content (~4 ppm). This material exhibits a low age uncertainty (0.2%, 2s; unanchored, ID-TIMS), allowing for the establishment of a well-defined isochron, particularly when anchored to the initial Pb ratio using LA-ICP-MS. ID-TIMS analyses of micro drilled C1 cement (17 sub-samples) produce a lower intercept age of 321.99 ± 0.65 Ma, an initial $^{207}Pb/^{206}Pb$

ratio of 0.8495 ± 0.0065, and a Mean Square of Weighted Deviations (MSWD) of 5.1. The systematic uncertainty of 1.5% observed in repeated LA-ICP-MS analyses challenges previous estimations of 2 – 2.5 % based on repeated analyses of ASH-15D and JT using WC-1 as primary reference material, underscoring the precision and reliability of RA138 for U–Pb dating applications.

# 1 Introduction

Recent years have seen a growing interest in the application of Laser Ablation Inductively Coupled Plasma Mass Spectrometry (LA-ICP-MS) to carbonate U–Pb geochronology. This method provided an accessible way to determine carbonate crystallisation age, depending partially on the availability of the well-characterised, matrix-matched calcite primary reference material (RM) WC-1 (Roberts et al., 2017). The establishment of an accepted methodology for corrections and age calculations (Coogan et al., 2016; Li et al., 2014; Ring and Gerdes, 2016; Roberts et al., 2017) has resulted in increasing number of

publications (Hoareau et al., 2021; Kylander-Clark, 2020; Rasbury et al., 2023; Giorno et al., 2022) spanning a diverse range of applications including paleoclimate (Chaldekas et al., 2022; Drake et al., 2017; Woodhead and Petrus, 2019), hydrothermal (Brigaud et al., 2020; Burisch et al., 2017; Macdonald et al., 2019; Mottram et al., 2020; Piccione et al., 2019), and tectonic



(Hansman et al., 2018; Looser et al., 2021; Nuriel et al., 2019; Nuriel et al., 2017; Weinberger et al., 2020) activities, and pedogenic and diagenesis processes (Elisha et al., 2021; Godeau et al., 2018; Mangenot et al., 2018; Methner et al., 2016; Scardia et al., 2019; Sindern et al., 2019).

Several additional RMs have been dated and distributed within the scientific community. However, their suitability as primary RMs is either limited due to their young age and low Pb count rates (Nuriel et al., 2021), or pose limitations due to their exhaustion (Guillong et al., 2020). Other published RMs, such as B6 (Pagel et al., 2018), lack independent age constraints from primary measurement techniques such as isotope dilution thermal ionization mass spectrometry (ID-TIMS). A comprehensive overview of available and proposed carbonate RMs can be found elsewhere (Wu et al., 2022).

While the WC1 is the most commonalty used primary RMs, it is not the most suitable, and the scientific community is still in search for a better reference material (Roberts et al., 2017). The significant uncertainty in the ID-Isotope Ratio Mass Spectrometry (IRMS) age of WC1 (254.4 ± 6.4 Ma) imposes a minimal uncertainty of > 2.5% on any obtained carbonate LA-ICPMS age. In addition, the very high MSWD of 1069 (when the isochron is anchored to the LA-ICP-MS initial $^{207}Pb/^{206}Pb$ ratio of 0.85 ± 0.04), suggest that WC1 is not homogenised and most likely exhibits natural heterogeneities. Consequently, the use of WC-1 as a primary reference material imposes constraints on achievable precision and accuracy.

As part of a collaborative effort to use carbonate U-Pb LA-ICP-MS geochronology of syn-depositional botryoidal cements as a stratigraphic tool (Samankassou et al., 2024), we dated several samples from carbonate debris flows sourced from a series of steep and high-relief carbonate platforms that developed during the Carboniferous in the marine foreland basin of the Cantabrian Zone (CZ) in northern Spain. Some of these samples have yielded excellent isochrons, suggesting that they may be suitable as natural RM for carbonate U-Pb dating by LA-ICP-MS. Here, we present a detailed characterisation of one of these samples, RA138, which besides its suitability for producing well-defined isochrons, was chosen because of the available amount of material (>6 kg), the ease of access to the location for any future sampling and the abundance of botryoidal cement in the outcrop samples which we identify as promising source for future RM for the growing LA-ICP-MS U–Pb geochronology community.

## 2 Sample

RA138 denotes that the sample was collected at meter 138 of section A, recorded at La Robla Quarry near the village of La Robla in northern Spain. The precise geographical coordinates are: 42.808355N and -5.687950E (EPSG:4258 (ETRS89)) respectively 42°48'29.84"N and 5°41'16.69"W (Degrees Minutes Seconds; DMS) Google Earth satellite image are. A substantial sample exceeding 6 kg in weight was extracted from a microbial boundstone lithoclast characterised by the presence of exceptionally well-preserved botryoidal cements. This lithoclast was embedded within a matrix-supported carbonate debris flow interbedded within uppermost Serpukhovian limestone strata. Comprehensive discussions regarding the samples and the regional geology can be found in Samankassou et al. (submitted). During the analytical process, several thin sections were prepared and several blocks measuring approximately 30x20x5 mm were cut for subsequent polishing, cold



cathodoluminescence (CL) imaging and the identification of botryoidal cement generations. Further details on selected individual blocks are available in the supplemnent figures S1-S5.

## 3 Methods

LA-ICP-MS stands as the primary methodology for which RA138 serves as the designated RM. Furthermore, LA-ICP-MS was employed for comprehensive characterisation, including analyses of trace element distribution and U-Pb age

determination. Independently, two laboratories utilized ID-TIMS analyses as a supplementary method to establish the reference U–Pb isotopic ratios.

In tandem with these analytical techniques, cathodoluminescence (CL) imaging was acquired using an Olympus polarizing microscope equipped with a CITL 8200 Mk5-2 stage, operated at 15 kV and 130 μA, and an Olympus DP74 camera was used. Combined with microscope images of polished sections, this approach revealed growth zoning and the different phases present

in the sample.

### 3.1. LA-ICP-MS

LA-ICP-MS analyses were conducted using a Resonetics (now Applied Spectra) Resolution 155HR LA system coupled to a Thermo Element XR single collector sector field ICP-MS. This setup included a high-capacity interface pump to improve sensitivity, along with the addition of small quantities of nitrogen (2ml min$^{-1}$) in the makeup argon gas flow (Wu et al., 2022).

The same RA138 sample section was analysed during a series of individual sessions (n=23), aiming to demonstrate repeatability and homogeneity of the material. The LA parameters employed for U-Pb dating were consistent with previous methodologies (Guillong et al., 2020), utilizing a 110 μm static spot ablation at 5 Hz and an energy density of 2 J cm$^{-2}$. Key parameters are concisely outlined in supplementary Table S1. Data reduction adhered to established procedures (Guillong et al., 2020; Roberts et al., 2017), with partial processing accomplished in Iolite 4 (Paton et al., 2011; Petrus and Kamber, 2012).

This involved the selection of integration intervals, gas blank subtraction, and downhole fractionation correction (Paton et al., 2010), utilizing WC-1 and the UcomPbin (Chew et al., 2014) data reduction scheme, without implementing common Pb correction on the RM.

Further data treatment occurred in Excel, where the downhole-corrected $^{206}Pb/^{238}U$ ratio, $^{207}Pb/^{206}Pb$ ratio, raw counts per second, and error correlation were exported from Iolite. This treatment encompassed drift correction for the $^{206}Pb/^{238}U$ ratio,

referencing NIST 614 and NIST 612, $^{207}Pb/^{206}Pb$ ratio calculation (ratio of mean), and normalization to the known NIST 614 ratio (0.8407 GeoReM preferred value (Jochum et al., 2005)). Corrected $^{207}Pb/^{206}Pb$ ratios and intermediate $^{238}U/^{206}Pb$ ratios were subsequently plotted using IsoplotR (Vermeesch, 2018) in a Tera–Wasserburg (TW) diagram, enabling the calculation of a Discordia model 1 anchored to a fixed initial Pb value (0.85 for WC-1 and 0.8495±0.0065 for RA138) to determine a lower intercept age.

To account for the known age of the primary reference material (RM), a correction factor was calculated and applied uniformly to all unknown $^{238}U/^{206}Pb$ ratios in each session, thus ensuring accuracy for the $^{238}U/^{206}Pb$ ratios of all spot analysis. The



accuracy of this data reduction methodology was rigorously tested across all sessions using multiple secondary RMs, typically JT (Guillong et al., 2020), B6 (Pagel et al., 2018), and ASH-15D (Nuriel et al., 2021). Measurement uncertainties arising from the ratios, as well as uncertainties associated with drift correction and the $^{238}U/^{206}Pb$ correction factor, were quadratically propagated.

In addition to the dating analyses, we conducted an in-depth investigation of trace elements within the various phases using a laser spot diameter of 30 μm. NIST 612 served as the reference material, and stoichiometric calcium (40 wt% Ca) was employed as an internal standard for ablation yield correction. Trace element maps were acquired in scanning mode to examine the distribution of both trace elements (TE) and U/Pb ratios.

For low resolution mapping purposes, a square spot size of 91 μm, a repetition rate of 10 Hz, an energy density of 2 J cm⁻², and a scanning rate of 50 μm s⁻¹ were employed. A total of 32 isotopes were measured with a sweep time of 0.585 seconds, and data reduction was performed using Iolite 4.8.3. NIST 614 was utilized as the reference material, and 40% weight of calcium (Ca) served as the internal standard. All maps are shown in the supplement figures S6 and S7.

For high resolution maps a Teledyne Iridia LA system equipped with a Cobalt fast washout ablation cell was coupled to an Agilent 8900 ICP-MS equipped with a quad lock system (Norris et al., 2021) at the National History Museum in London. A square spot size of 20 μm, a repetition rate of 298.1 Hz, an energy density of 2 J cm⁻², and a scanning rate of 424.3 μm s⁻¹ were employed. Nine masses (Mg24, Ca43, Ti47, Mn55, Fe57, Sr86, Y89, Pb206 and U238) were analysed with a sweep time of 47 ms. Washout was tested to be < 5 ms to 10 % of the maximum signal from a single laser pulse. Data was processed and maps were constructed using Iolite 4(Paton et al., 2011).

**3.2 ID-TIMS**

RA138 was analysed by isotope dilution thermal ionization mass spectrometry (ID-TIMS) in two sessions. All samples were purified at the University of Geneva and subsequently analysed by TIMS either in Geneva (session 1) or at both Geneva and ETH (session 2). The botryoidal cement C1 was drilled to produce 1–4 mg fine powder from polished rock surface using a Dremel drill with drill bit diameter of 0.8 mm, allowing sample spots with approximately 1 mm diameter. The sample powders were collected in 12 ml PMP beakers, dissolved in 100 μl 6 M HCl and equilibrated with 5–9 mg of the EARTHTIME $^{205}Pb$–$^{233}U$–$^{235}U$ (ET535) tracer solution(Condon et al., 2015; Mclean et al., 2015)  during 30 min on an 80 °C hotplate. The solutions were then dried down and re-dissolved in 1 M HBr. Pb was purified in an HBr ion exchange chemistry using 50 μl of AG1-X8 anion resin (Cl-form, 200–400 μm mesh). The U cuts were converted to chloride form, then purified through a second pass on the AG1-X8 columns (rinsed in 3 M and 6 M HCl, eluted in water). Due to the high cation concentration in carbonates, the U fractions were then further put through a separate set of 50 μl columns filled with RE resin (TRISKEM B50-S, 50–100 μm). Pb and U fractions were dried down with trace $H_3PO_4$ (0.02M) in separate 7 ml PFA beakers.

Uranium and Pb isotopic ratios were analysed on two thermal ionisation mass spectrometers (TIMS): a Thermo Triton at the University of Geneva and a Thermo Triton Plus at ETH Zurich, both equipped with $10^{13}$ Ω amplifiers. U and Pb aliquots were



loaded at the respective labs on separate, zone-refined, outgassed Re filaments in 1µl of silica gel emitter (modified after (Gerstenberger and Haase, 1997)). In both cases, Pb isotope measurements were done in static mode: either using Faraday cups for all Pb masses (ETH) or combining Faradays for $^{205-208}$Pb with an axial SEM for $^{204}$Pb (Geneva) (Von Quadt et al., 2016). Instrumental mass fractionation of Pb isotopes was corrected with factors specific to each detector setup and derived from a compilation of mass fractionation factors measured in Pb isotopic standards and double-spiked unknowns. U was

measured as $UO_2$ in static mode, with oxide interferences corrected either using in-run $^{18}O/^{16}O$ determined with mass 272 (Wotzlaw et al., 2017), or assuming a value of $0.00205 \pm 0.00004$. Mass fractionation was corrected with the known spike $^{233}U/^{235}U$ value of 0.9950621 and assuming a sample $^{238}U/^{235}U$ ratio of $137.818 \pm 0.045$ (2σ) (Hiess et al., 2012).

    Raw data were evaluated using the Tripoli software (Bowring et al., 2011) and reduced using the algorithms of (Schmitz and Schoene, 2007). Based on total procedural blank measurements at the University of Geneva, 3.5 pg of common Pb was assigned

to laboratory blank and corrected with its long-term isotopic composition. Sample ages were evaluated using the 2D Tera-Wasserburg diagram in IsoplotR (Vermeesch, 2018) using the decay constants of (Jaffey et al., 1971). Considering the old age result, the U/Pb ratios were not corrected for initial Th, Pa, or U disequilibrium.

## 4 Results

### 4.1 Textures

Sample RA138 exhibits at least four distinct phases of precipitation, as illustrated in Figure 1a (visual image), Figure 1b (cold cathodoluminescence light, CL, image) and Figure 4 (LA-ICP-MS compositional maps):

1.    Matrix: Visually and under CL, the matrix forms a patchy appearance and consists of clotted peloidal micrite with scarce bioclasts. Mm-sized internal porosities are lined by thin isopachous rims of fibrous cements (<100 μm thick)

150        and incipient botryoidal cement fans. Remaining porosity is occluded by blocky calcite cements.

2.    Cement C1 (botryoidal): This phase is characterized by a dark colour and a fibrous appearance observed in sections with concentric growth bands perpendicular to the direction of cement growth. Under CL, it appears dark (low or no luminescence) and relatively homogeneous.

3.    Replacement Cement C2: This phase replaces Cement C1 and displays a fine-grained texture with a brighter

155        appearance, both visually and under CL (high or medium luminescence).

4.    Radiaxial Fibrous and Sparry Cement RFSC3: This phase exhibits a white to transparent appearance and very bright under CL (high luminescence). It is distinguished by the larger crystals exceeding 200 μm in size.

### 4.2 LA-ICP-MS Trace elements and distribution

    Trace elements were quantified in the four distinct phases through segmentation (i.e., selection of region of interests) of a LA-

ICP-MS map. Additionally, single spot analyses were performed on the C1 botryoidal cement. A comprehensive presentation



of all data is provided in the supplementary Table S5. Selected elements are highlighted for comparative analysis among the phases in Figure 1d.

Notably, the C1 botryoidal cement, suggested as the potential new RM exhibits significantly lower concentrations of Mg, Mn, and Fe compared to all other phases and higher levels of U and Sr compared to C2 and RFSC3 phases. The C1 phase is

identified best by this set of elements. In the matrix, U and Sr concentrations are comparable, but all elements show a more heterogeneous distribution. Uranium concentration in C1 is remarkably homogeneous, falling within the range of 4 µg g$^{-1}$ with a precision as Relative Standard Deviation (RSD) of less than 10%.

Conversely, the replacement Cement C2 displays elevated concentrations of Mg, Mn, and Fe. Owing to its lower U content and higher initial Pb, spot analysis within this phase plot closer to the upper intercept in the Tera Wasserburg (TW) diagram

(Figure 1c). This phase exhibits a broader spread than the isochron defined with the Cement C1 only.

Additional high-resolution TE maps (Figure 4) reveal the fibrous structure of C1 cement in the Y and U content and show some isolated, high Mg spots in C1 cement. The higher spatial resolution imaging also captures details like the decreasing Y content towards the rim of the C1 cement, which are not detected on our low-resolution imaging (supplementary figure S6).

## 4.3 LA-ICP-MS U-Pb Dating

Cement C1 was dated during 23 sessions (November 2022-July 2023), using WC-1 as the primary reference material for $^{238}$U/$^{206}$Pb ratio correction. A detailed dataset is provided in Table S2a in the supplementary. Approximately 5% of the RA138 analyses were defined as outliers based on their deviation from the isochron. This divergence is likely attributed to the ablation of different phases other than C1 or mixing of C1 with other phases (i.e., misplacing ablation spots or ablating other phases in

lower parts of ablation pits), notably observed as bright spots within C1, as visualized in CL images (Figure 1b). When analysing all different phases (C1, C2, RSFC3 and the matrix, Figure 1c), the number of points deviating from the isochron in the TW space increases, some uncertainty ellipses are bigger due to low U concentration, the age becomes younger and the initial Pb composition can be different.

The pooled TW isochron, comprising n=763 spot analyses (Figure 2a and data in Table S2b), reveals a lower intercept age of

319.25 ± 0.48 Ma (without propagation of excess uncertainty S$_{sys}$), with a Mean Square of Weighted Deviations (MSWD) of 0.81, and an initial $^{207}$Pb/$^{206}$Pb ratio of 0.8486 ± 0.0016 (unanchored). Figure 2b shows a representative session with n=30, producing a lower intercept age of 322.69 ± 2.09 Ma (without propagation of excess uncertainty S$_{sys}$), an MSWD of 0.37, and an initial $^{207}$Pb/$^{206}$Pb ratio of 0.8495 ± 0.0051, anchored to the ID-TIMS value of 0.8495 ± 0.0065.

The 23 lower intercept ages span from 314.26 ± 2.25 to 323.83 ± 3.23 Ma, with MSWD values ranging from 0.12 to 2.0 (See

Table S2a for details). The calculated weighted mean of these lower-intercepts ages of all 23 sessions yields an age of 319.25 ± 1.33 Ma (0.4%, without propagation of excess uncertainty S$_{sys}$) with an MSWD of 7.6 (Figure 2c and data and calculations in Table S3). To address potential systematic uncertainties, a quadratic propagation of a 1.5% uncertainty to the 23 individual intercept ages results in a comparable weighted mean age of 319.32 ± 1.1 (0.34%, without propagation of excess uncertainty S$_{sys}$



$S_{sys}$), with a more acceptable MSWD of 1.3 (Figure 2d). All computations were conducted using IsoplotR, and model 1
discordia isochrons, with 95% confidence interval uncertainties.

### 4.4 ID-TIMS U-Pb dating

We present a set of 21 ID-TIMS analyses conducted on botryoidal cement C1 during two distinct measurement campaigns.
The initial campaign (n=9) took place at the University of Geneva, while the subsequent campaign involved a collaborative
effort between the laboratories of ETH Zürich (n=8) and Geneva (n=4). Detailed ID-TIMS data are provided in supplementary
table S4, and the complete dataset of 21 analyses is shown in Figure 3. The resulting isochron of all analyses exhibits a lower
intercept age of 320.76 ± 1.31 Ma, an initial $^{207}Pb/^{206}Pb$ ratio of 0.8378 ± 0.0102, and a significant overdispersion reflected in
an MSWD of 76.

Notably, the four analyses with the highest $^{207}Pb/^{206}Pb$ ratio are distant from the isochron constructed with the remaining data,
indicating a potential mixing with non-C1 phase material which may be characterised by either a distinct common Pb
composition or age (Fig. 1c). Examination of the sampling locations during the second campaign, as illustrated in the CL image
(Figure 3c), suggests the possibility of mixtures of the micro-drilled sample due to the depth of pit required to obtain sufficient
material for the ID-TIMS analyses. Excluding these four points from the age interpretation (Fig. 3) yields a preferred lower
intercept age of 321.99 ± 0.65, an initial $^{207}Pb/^{206}Pb$ ratio of 0.8495 ± 0.0065, and an MSWD of 5.1. The elevated MSWD
value indicates the presence of unresolved sample heterogeneity due to the large sampling volume, mixed sampling of other
phases than C1 seems possible. These refined results, including the overdispersion term (Vermeesch, 2018) for both intercepts,
are considered the preferred values for utilization as a reference in LA-ICP-MS work.

### 5 Discussion

Botryoidal cement C1 within sample RA138 displays a favourable age homogeneity and a large spread of U/Pb ratios,
rendering it a suitable primary RM for the $^{238}U/^{206}Pb$ correction in carbonate LA-ICP-MS U-Pb dating. The 0.2% age
uncertainty from the ID-TIMS analyses enhances the precision of the correction factor and consequently improves the age
uncertainty of unknown samples. This stands in contrast to WC-1 which has a higher uncertainty of 2.5% (254.4 ± 6.4 Ma)
(Roberts et al., 2017).

Furthermore, a re-evaluation of the excess uncertainty ($S_{sys}$) previously estimated for U-Pb analysis by LA-ICP-MS,
particularly in zircons (Horstwood et al., 2016), indicates approximately 1.5% excess uncertainty ($S_{sys}$) based on repeated
analyses of RA138 (Figure 2b and 2c), challenging the previously assumed range of 2–2.5% (Guillong et al., 2020). RA138
thus provides an improved reference material with less scatter around the U/Pb isochron as a prerequisite for improving this
method.



While RA138 contributes to enhanced precision, it is essential to acknowledge its lack of homogeneity. Apart from C1, other
phases are present. Precise selection of analysis locations for calculating correction factors is crucial, avoiding less suitable
locations such as the matrix or replacement cements C2 and RFSC3. C2, for instance, exhibits higher initial Pb than C1, with
some analyses aligning with C1, but a substantial portion revealing open-system behaviour and younger crystallisation ages
(Figure 1c). Radiaxial Fibrous and Sparry Cement RFSC3, containing less U, demonstrates an overall younger age with much
higher variability (Figure 1c).

The matrix, although visually heterogeneous and displaying a broad range of U-Pb ratios, is not recommended as RM due to
potential open-system behaviour and some younger ages. Even when analysing only C1, up to 5% of point analyses may be
outliers, likely arising from the partial analysis of small parts of C2, identifiable as small bright spots in the CL image (Fig.
1b). Therefore, it is suggested to conduct approximately 30 analyses per session for a robust correction factor calculation. It is
also recommended to image well-characterised RMs sections used in each laboratory (e.g., petrography, trace elements
screening) for better visualisation and assessment of the different phases.

While C1 of RA138 exhibits a larger variation in U/Pb than other commonly used RMs, and unanchored data typically have
an upper intercept within the uncertainty of the ID-TIMS initial $^{207}$Pb/$^{206}$Pb ratio, it is still recommended to anchor the isochron
for correction factor calculation to the ID-TIMS-determined initial $^{207}$Pb/$^{206}$Pb ratio of 0.8495 ± 0.0065, given the higher
precision and reduced spread in intercept ages.

**6 Conclusions, availability, and outlook**

Botryoidal cement C1 from RA138 provides a significant potential for reducing uncertainties related to primary reference
material corrections during U-Pb LA-ICP-MS in carbonate dating. Its variable U/Pb ratios, coupled with a relatively high and
homogenous U content, contribute to a reasonably low uncertainty, resulting in a well-defined isochron, particularly when
anchored to the initial Pb ratio. This material is sourced from an accessible outcrop in northern Spain ensuring unlimited supply
and numerous aliquots have been prepared for distribution. Detailed information on aliquots is included in the supplementary
figures S1-S5 and interested parties can request additional details from the corresponding author. With the dissemination of
this new RM, it is expected that achievable precision in LA-ICP-MS calcite dating will substantially improve. This is a crucial
step forward, while the community is searching for RMs that would be homogenous not only in age but also in U/Pb ratio – to
be used directly for all corrections making data reduction and downhole fractionation correction significantly easier.


**7 Author contribution**

MG, IM, ES, and MO planned the campaign, OMT and JRB collected the samples, ES, LT, NL and MG prepared the samples,
ES did the petrography, MG, LT and NL did LA-ICP-MS U-Pb Analyses, MG, LT, DS and YB did LA-ICP-MS mapping



analyses, IM and DS did ID-TIMS analyses, MG, LT and NL did the CL imaging, MG wrote the manuscript draft. All authors
reviewed and edited the manuscript.

## 8 Competing interests

The authors declare that they have no conflict of interest.

## 9 Acknowledgements

Critical reading and comments by Perach Nuriel improved the manuscript.
Funding from Swiss National Science Foundation (Grants 200021_160019, 206021_133771, 200021_169849 and
200021_182556), Spanish Government (CGL2013-44458-P, PGC2018-099698-B-I00 and PRX19/00423), the Principality of
Asturias (AYUD/2021/51293) and FEDER is acknowledged. DS and LT were partly supported by an ETH Zurich Career Seed
Award.



Figures:



**Figure 1: Characterisation of the different phases found in RA138: a: microscope image with 4 different phases as well as spot location in 3 phases shown in c. b: CL image of the square region of a with spot locations c: Tera Wasserburg diagram of the spots shown in a and b and matrix analyses. D: results of trace element concentration analysed in the 4 different phases.**






**Figure 2: LA-ICP-MS U-Pb results for RA138 using WC-1 as RM. a: pooled isochron from 23 sessions b: typical results from one analytical session c: rank order plot of isochron ages and weighted mean calculation for 23 sessions showing excess scatter with an MSWD of 7.6 d. same data as c but including a systematic uncertainty of 1.5 % resulting in an MSWD of 1.3.**






**Figure 3: ID-TIMS results for RA138: Preferred isochron representing cement C1 in sample RA138 to be used as reference values in blue. All results, including higher $^{207}$Pb/$^{206}$Pb ratios that are omitted for the preferred isochron labelled with an arrow in black. Insert: CL image before ID-TIMS sampling including the locations.**








**Figure 4: CL image and trace element maps of RA138. A: CL image b: Mg concentration map showing the low Mg content of C1 compared to other regions. d: Y concentration map d: U concentration map showing higher content in the C1 cement.**



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
