# Peer review of "Technical note: RA138 Calcite U-Pb LA-ICP-MS primary reference material"

_Geochronology, 2024_

## Author Response (AR1)

Point by point reply to the comments.

RC1:

This paper by Guillong et al. advances an important goal of characterizing a primary reference standard for the calcite U-Pb isotopes. It is well written and well documented, but I do have a few minor criticisms for the authors to consider:

1. In line 31, please consider inserting Gulbranson et al., 2022 ( https://doi.org/10.3390/geosciences12090346 ) in this list of appropriate citations. Also include it at the end of the list of citations in line 35)
2. In line 48, please clarify the status of Samankassou et al. (2024). Is this also referred to as "submitted" as in line 63?
3. As presently shown the cathodoluminescence image in Figure 1b is too dark to see the difference in luminescence colors. I do understand the viewpoint that original images should be shown as collected but given the reality of dim luminescence in many geological samples, I recommend increasing the brightness and contrast of the image so that it conveys appropriate information. There are many other things that operators can do to achieve this end in original native photomicrographs, such as increasing accelerating voltage, beam current, or streaming helium into the chamber. Given all of these possible variables, I suggest simply doing some digital processing of the image to clearly show the features of interest.
4. Great job on an important contribution!

Reply:

We would like to thank Greg Ludvigson for the comments.

1. We will add the requested literature Gulbranson et al., 2022 into the revised manuscript.
2. Yes, this is the same manuscript we cited, and we will update the status on submission of the revised manuscript.
3. We agree that the CL image in Figure 1b is too dark and will enhance the image quality for a revised version of the manuscript so that more information becomes visible. As this is a stitched image of about 30 individual images, we suspect that the stitching software decreased the brightness to make it balanced, and we did not correct it. An image enhancement is no problem.
4. Thank you.

RC2:

General comments

Carbonate U–Pb geochronology is increasingly important for various research fields in geosciences, and lack of high-quality reference materials (RMs) are critical issue for acquiring reliable age data. Owing to the limited numbers and the poor quality of RMs, applications of carbonate U–Pb geochronology can be retarded. The RM138 presented in the manuscript is well-characterised and demonstrating the better homogeneity in terms of the U–Pb age compared to previously reported carbonate RMs. Although I believe that this manuscript should be of interest to the audience of GChron, and be suitable for publication, there remains several questions and points should be addressed.

Specific comments

L13: As for terminology, "U–Pb" (en dash) rather than "U-Pb" (hyphen) is recommended for describing the relationship between U and Pb as parent and descendent isotopes.

L43: I would like to recommend using either LA-ICP-MS or LA-ICPMS as a consistent abbreviation for laser ablation ICP mass spectrometry through the manuscript.

L59: Where in the manuscript is the google satellite image shown?

L77: Although the authors describe the correction scheme for U–Pb isotopic data obtained by LA-ICP-MS in detail, the actual values for key correction parameters, such as the relative sensitivity ratio of U and Pb, mass bias factors, and down-hole fractionation, are not stated. In objectively assessing the data quality, I would like to suggest that these values are shown in the manuscript.

L112: The notation for isotopes should be changed from Mg24 to $^{24}$Mg.

L137: In the manuscript, the $^{235}$U/$^{238}$U value of the sample is assumed to be 1/137.818 for the calculation of the mass bias factor as a representative value for the natural uranium isotopic ratio. The value of 1/137.818 was previously determined from zircon and apatite reported by Hiess and co-authors, and the value may not necessarily apply to carbonates. In fact, the $^{235}$U/$^{238}$U value of marine carbonates deviates from the value for zircon and apatite, and some carbonates can demonstrate fractionated $^{235}$U/$^{238}$U potentially depending on redox conditions. For carbonates, the degree of potential isotopic fractionation for $^{235}$U/$^{238}$U is within 0.1%, but this can be a cause of significant systematic error for high-precision U–Pb isotopic analysis based on ID-TIMS. Although quantitative evaluation for the systematic error may be difficult without measuring the actual $^{235}$U/$^{238}$U isotope ratios, I would like to recommend mentioning the potential systematic error arising from the assumption of the natural U isotopic ratio in the manuscript.

L204: In Fig. 3, the treatment of the ID-TIMS data points for the D16 domain in the C1 cement with high non-radiogenic Pb contents can appear arbitrary. The authors indicate that there is a contribution from non-C1 phases, but, for a clear rationale, showing some evidence for containing non-C1 components within the D16 domain is preferable. For instance, some elements enriched in non-C1 phases (e.g., Mg, Mn, and Fe) should be also

high for the aliquots of the D16 domain. In addition, if there are any magnified photographs of the D16 domain before isotope analysis, I would like to recommend including them in the manuscript.

Fig 2. B: For easy understanding from readers, I would like to recommend demonstrating the intercept point of the regression line and the Concordia curve on the Tera-Wasserburg diagram.

Reply:

We Thank Niki Sota for the detailed review and the valuable comments that will help to improve the manuscript.

L13: We agree that the U–Pb terminology should be consistent throughout the manuscript, and we will make sure the revised manuscript, as suggested, will use the dash (–) instead of the minus (-).

L43: We agree that the abbreviation of Laser ablation inductively coupled plasma mass spectrometry (LA-ICP-MS) should be consistent throughout the manuscript and we will adjust it.

L59: There is no Google Earth image and we do not intend to add a Google Map / Earth Image as everyone can enter the coordinates and look at the location directly in Google Earth. Additionally, the Google satellite image is not perfectly accurate to the reel coordinates and might change in future. That is also why there are two set of coordinates, the "real" ones in the (EPSG:4258 (ETRS89) and the ones for Google Earth/Maps Image that shows the location on the satellite image. To prevent any confusion, we will change the revised manuscript and just report the real coordinates.

L77: The correction factors for the 23 sessions can be added but we think that this information is of limited help. Specifically:

1) The mass bias for the 207/206 correction is found to be very small and stable for different sessions (usually less than 1% variation).
2) The correction factor for 238U/206Pb is highly variable due to variations in ionisation efficiency and strongly depends on the daily tuning of the ICP-MS instrument. In Wu S et al., 2022, as an example, the correction factors are given and vary between 1.063 and 1.189 for 25 sessions. We observed similar values, but other instruments might have different values and different ranges.
3) The correct assessment of downhole fractionation using calcite is difficult due to variable initial Pb content, possible surface Pb contamination, and the generally small amount of change in the ratio due to the low drill rate and large craters. Again, an example is given in Wu S et al., 2022 for WC-1, and we observed similar results.

The present manuscript focuses on the characterisation of RA-138 with ID-TIMS and LA-ICP-MS including the repeatability and does not specifically deal with the data reduction method and matrix effects. This was already described in Guillong et al., 2020 and Wu S et al., 2022 and we do not think repeating this content would add value to this manuscript.

L112: We agree and will change in the revised version of the manuscript.

L137: We acknowledge that the choice of $^{238}U/^{235}U$ measured in accessory minerals by Hiess et al., 2012 is not ideal. This value corresponds to $\delta^{238}U$ = -0.19 +/- 0.3 ‰ (2 sigma, relative to CRM 112a) and comes with a large uncertainty which is propagated into every date. We did not

measure $^{238}U/^{235}U$ in our aliquots and have no independent knowledge of the appropriate value; however, compilations of $\delta^{238}U$ in modern and ancient marine carbonates show significant variability in natural systems. For example, Chen et al., 2021 compiled data where Phanerozoic carbonates are on average $\delta^{238}U$ = -0.37 ‰, with a significant spread of ca. 0.6 ‰ (2 SD). We do not know whether this could be random (i.e. compositions vary for each of our aliquots) but suspect that U isotopic composition should be coherent at least within the same cement generation. As such, a deviation of $\delta^{238}U$ from the Hiess et al., 2012 value should result in a systematic error, as suggested by the reviewer.

The magnitude of this error can be estimated. Assuming that RA138 has $\delta^{238}U$ = -0.37 of average Phanerozoic carbonate instead of -0.19 results in a negligible shift of the isochron age of 10 ka (30 ppm relative) towards younger values and has no effect on the value of the common Pb intercept. As such, in this particular case and at the currently achievable level of precision in ID-TIMS, the exact $^{238}U/^{235}U$ of the carbonate can be neglected.

We note, however, that future studies employing high-precision U-Pb geochronology of (particularly old) carbonates should consider directly analysing $^{238}U/^{235}U$ in the unknowns to establish this value. In younger samples, estimates of initial $^{234}U/^{238}U$ disequilibria are likely to be the dominant limitation to accuracy; however, in developing RA138 as a reference material we are interested in the raw date before any disequilibrium corrections rather than its true age.

X. Chen, F. L. H. Tissot, M. F. Jansen, A. Bekker, C. X. Liu, N. X. Nie, G. P. Halverson, J. Veizer, N. Dauphas, The uranium isotopic record of shales and carbonates through geologic time. Geochimica et Cosmochimica Acta 300, 164-191 (2021).

L204: We agree that for the ID-TIMS results showing some clear evidence for containing non-C1 components within the D16 domain would be preferable, however this is difficult to achieve as all the solutions have either been used or were disposed so a direct analysis for elevated Mg, Mn and Fe is not possible. A high-resolution image of the sample prior to the micro drill sampling does not exist and would not help as we selected the sampling location based on the CL image (high resolution) presented in Figure 3 and we (obviously) target the reliable domains. However, the CL image is from the surface only, and the micro drill sampling removes several 100 to 1000s of micrometres into the sample. On the CL image there is a partial cut from the sawing between sampling locations 4 and 6 revealing a finite depth of the C1 phase. Based on this observation we think it is valid to suggest that the micro drilling may have touched a different phase. Another CL image would reveal this, but unfortunately the sample has been used for other experiments and extensive micro drilling.

Fig 2. B: We will extend the x scale to include the intercept of the regression with the Concordia curve for the revised version of the manuscript

Editor Comment:

One minor comment, possibly irrelevant for the purpose of this study (reference material). You state the sample comes from a Serpukhovian section, but the age you get is Bashkirian. Is this due to late formation of the carbonate (geological process) or an unrecognized bias in the analytical method

Reply:

We thank the Editor for bringing this inconsistency up with the analysed age not being in the suggested strata.

The authors are indeed aware and had a debate some time ago concerning the age provided by ID-TIMS. In that debate we commented on the discrepancy of the ID-TIMS age obtained for the botryoidal cement of sample RA-138 and the expected age of deposition of the debris flow strata within the measured section.

According to the available biostratigraphic information the age of the debris flow strata is uppermost Serphukovian as the strata lays below the FAD of the conodont D. inaequalis index for the Mississippian–Pennsylvanian boundary and the FAD of Pennsylvanian (Bashkirian foraminifera). An age of ca. 323.8 ± 0.4 Ma is expected for both the debris flow strata and botryoidal cements based on our well-constrained correlation.

One of the potential causes of this very small discrepancy (about 0.5% of the obtained age) could maybe be related to post-depositional processes affecting the botryoidal cements during the burial history of the sample. In this concern, it is relevant to mention that despite the exceptional preservation of the original fabrics of botryoidal cements, and both the geochemical data and CL shown in the manuscript (suggesting and exceptional degree of preservation of the botryoidal cements), as revealed by clumped isotopes performed in another sample RA-137.5 (collected from the same debris flow strata) confirms that the sediments were deeply buried (probably as deep as 4–6 km according the temperature data inferred from clumped isotopes and the vitrinite reflectance measured in Tournaisian black shales in the same area of study). Oxygen isotopes and Sr contends reveal that the diagenesis took place under a rock-buffered conditions with a limited interaction with diagenetic fluids, but maybe the effects of diagenesis could had played a role. The micro-Raman analyses performed in sample RA-137.5 show that although some remains of aragonite still occur in the samples, most of the current botryoidal cements were recrystallized/replaced by low-Mg calcite (Samankassou et al., in preparation). We certainly do not know if the original ratios in between U and Pb isotopes could had slightly varied during calcite recrystallization/replacement processes because of a fractionation/exchange with diagenetic fluids. It could be possible that the ID-TIMS age obtained for the sample (the age of the current calcite replacing the original aragonite) would record the age of calcite replacement/recrystallization instead of the age of precipitation of the original aragonite cements from seawater. This rather younger ages than expected was also observed by LA-ICP-MS in other samples and is part of the cited manuscript (Samankassou et al., in preparation).

Another possible explanation for a systematic offset of the ID-TIMS age is a possible 234U/238U disequilibrium in the precipitating fluid.

However, all these possibilities cannot be discussed in the present manuscript describing a reference material. The discussion will be part of another manuscript.

Other changes:

Line 92: we corrected the $^{207}Pb/^{206}Pb$ ratio of NIST 614 to the correct value of 0.8704.

We changed the order of Figures to match the appearance in the text: the former Figure 4 is now Figure 2 , 2->3 and 3->4 and we updated the text accordingly.

Some minor wording throughout the text was improved.

The reference Nuriel et al., 2019 was changed to refer to the correct paper.